# Dietary Fatty Acid Composition Impacts the Fatty Acid Profiles of Different Regions of the Bovine Brain

**DOI:** 10.3390/ani12192696

**Published:** 2022-10-07

**Authors:** Daniel C. Rule, Emily A. Melson, Brenda M. Alexander, Travis E. Brown

**Affiliations:** 1Department of Animal Science, University of Wyoming, Laramie, WY 82071, USA; 2Department of Integrative Physiology and Neuroscience, Washington State University, Pullman, WA 99164, USA

**Keywords:** bovine, DHA, EPA, arachidonic acid, fish oil, palm oil, calcium salts

## Abstract

**Simple Summary:**

The fatty acid composition of the mammalian brains varies by functional region, development, sex, and diet. In the current study, diets for growing female and castrated male bovines were supplemented with the calcium salts of palm or fish oil for 220 days. The cattle had unrestricted access to the supplements. At slaughter, the brains were removed from the cranium and blocked to the functional region. Analysis of the fatty acid content of the brain revealed they had similar fatty acid contents to other species including rodents and wild ungulates. Although diet differences across the regions were small, cattle supplemented with fish oil had greater contents of eicosapentaenoic acid (EPA) and docosahexaenoic acid (DHA), with arachidonic acid levels being greater in palm-oil-supplemented cattle. Fatty acid contents across regions were similar to what has been reported for other species, and it suggests that a conserved lipid metabolism creates a uniform brain fatty acid profile.

**Abstract:**

Fatty acid composition across functional brain regions was determined in bovine brains collected from cattle that were provided supplements of calcium salts containing either palm or fish oil. The Angus cattle were divided into two groups, with one group offered the supplement of calcium salts of palm oil and the other offered the calcium salts of fish oil (n = 5 females and n = 5 males/supplement) for 220 days. These supplements to the basal forage diet were provided ad libitum as a suspension in dried molasses. The fish oil exclusively provided eicosapentaenoic acid (EPA, C20:5 n-3) and docosahexaenoic acid (DHA, C22:6 n-3). The functional regions were dissected from the entire brains following commercial harvest. While the cattle provided diets supplemented with the calcium salts of palm oil had increased (*p* < 0.01) liver concentrations of C18:1 n-9, C18:2 n-6, and arachidonic acid, the fish-oil-supplemented cattle had greater (*p* < 0.01) concentrations of liver EPA, DHA, and C18:3 n-3. In the brain, DHA was the most abundant polyunsaturated fatty acid. In the amygdala, pons, frontal lobe, internal capsule, and sensory cortex, DHA concentrations were greater (*p* < 0.05) in the brains of the cattle fed fish oil. Differences among the supplements were small, indicating that brain DHA content is resistant to dietary change. Arachidonic acid and C22:4 n-6 concentrations were greater across the regions for the palm-oil-supplemented cattle. EPA and C22:5 n-3 concentrations were low, but they were greater across the regions for the cattle fed fish oil. The effects of sex were inconsistent. The fatty acid profiles of the brain regions differed by diet, but they were similar to the contents reported for other species.

## 1. Introduction

The fatty acid composition of the mammalian brain varies by functional region, development, sex, and the animal’s diet [1,2,3,4]. The factors affecting the concentrations of docosahexaenoic acid (DHA; C22:6 n-3) in brain tissue are of recent interest. DHA is required for normal fetal brain development, with deficiencies associated with attention deficit disorders [5].

Many studies have focused on the maintenance of appropriate DHA for continuous optimal functioning in an adult animal. Dietary n-3 fatty acids, primarily α-linolenic acid (ALA; C18:3 n-3) and DHA, influence the DHA content in the brain, with different functional regions of the brain varying in responsiveness to dietary supplements [1,6,7]. The conversion of ALA to DHA and of linoleic acid (C18:2 n-6) to arachidonic acid (AA; C20:4 n-6) does not occur in the brain sufficiently enough to support brain DHA and AA requirements [8,9]. The liver’s production of DHA and AA adequately accounts for DHA and AA brain consumption [8,10]. Eicosapentaenoic (EPA; C20:5 n-3), another n-3 fatty acid, is not known to accumulate in the brain due to the metabolism of this fatty acid in brain cells [11,12].

Fatty acid uptake by the brain requires transport through the blood–brain barrier (BBB). During development, DHA crosses the BBB by simple diffusion [13] and facilitated transport mechanisms [14]. The mRNA for membrane and intracellular fatty acid transport proteins were appreciably expressed in the BBB of mice during a long chain poly-unsaturated fatty acid (LC-PUFA) accretion period [15]. Beyond developmental stages of brain growth and into the adult brain, dietary fatty acids affect fatty acid composition in functional regions [6].

Female gonadal hormones have inconsistent effects on the fatty acid compositions of whole rat brains [16]. In eight brain regions, DHA did not differ among the cycling and the control rats that had not begun reproductive cycling [17]. Much of the DHA from dietary ALA is due to liver synthesis and is greater in females than in males due to the increased gene expression of desaturase [18]. Female rats supplemented with ALA had lower DHA concentrations in the hippocampus than males, but they had greater DHA contents in the prefrontal cortex, hypothalamus, and midbrain [19]. Low levels of dietary ALA increased brain DHA in estrogen-treated, ovariectomized rats compared with untreated ovariectomized rats. When ALA was supplemented to ovariectomized rats treated with estrogen, DHA concentration was greater in the hippocampus compared with untreated ovariectomized rats [19].

Dietary DHA influenced brain fatty acid content in baboons [20] but not in poultry [21]. Reports on the brain fatty acids of ruminant species are sparse. Whole brain homogenates of wild deer, elk, and pronghorn antelope had similar fatty acid compositions [22] to that of the average of 11 regions of mouse brains [1]. The pre-gastric microbiome of ruminant animals metabolizes most polyunsaturated fatty acids before small intestinal absorption [23] through biohydrogenation, where unsaturated fatty acids are hydrogenated via ruminal bacteria [24]. Due to biohydrogenation, relatively low DHA levels in ruminant brains would be expected. However, enough ALA is consumed in forage diets, characteristic of these animals, to support liver DHA synthesis [25] and brain uptake [22]. The brain lipids of domestic ruminants, including phospholipids and fatty acid profiles, have been reported [26], but a relationship between the fatty acid profile and the functional brain region remains to be determined. The objective of the current study was to characterize the fatty acid composition of the bovine brain in response to dietary supplementation and determine the influence of gonadal sex.

## 2. Materials and Methods

### 2.1. Animals and Diets

All animal procedures were approved by the University of Wyoming Institutional Animal Care and Use Committee (#08062013DR0008-03). Twenty randomly selected Angus-bred cattle consisting of 10 nulliparous females (heifers) and 10 castrated males (steers) were placed on forage-based diets and supplemented with the calcium salts of fatty acids of either palm (n = 5 heifers; n = 5 steers) or fish (n = 5 heifers; n = 5 steers) oil (Virtis Nutrition, Corcoran, CA, USA). Ad libitum access to the fatty acid supplements was provided by dried molasses lick tubs (Ridley Block, Whitewood, SD, USA; Table 1). The supplemental tubs contained approximately 113 kg of dried molasses with 30% by weight of fatty acid calcium salts. The basal diet consisted of harvested forage (hay) composed of alfalfa, brome grass, and wheat grass (15% CP). The cattle were group-fed with supplements provided at the pen, with the fish- and palm-oil-supplemented cattle penned separately. The cattle had ad libitum access to water and were provided with free choice trace mineralized salt blocks.

The cattle (approximately 10 months of age; 235.3 ± 0.2 kg heifers; 267.3 ± 3.3 kg steers, beginning weight) remained on this feed for 220 days. Cattle were harvested at approximately 18 months of age by commercial slaughter at the University of Wyoming Meats Laboratory abattoir with ending body weights of 398.4 ± 7.1 kg or 440.2 ± 7.1 kg for heifers and steers, respectively.

### 2.2. Brain Dissection

The cattle were killed using captive bolt and their heads were removed at the atlas joint for the required health inspection. The skulls were opened using a power saw, with the unfixed brains removed from the cranium within 10 min of exsanguination. The brain tissue was removed whole and retained its shape and landmarks in spite of the captive bolt damage. The studied areas were dissected fresh using the surface landmarks described in the atlas of Vanderwolf and Cooley [27], quickly placed in 1.5 mL Eppendorf tubes, and frozen on dry ice. The cortex was sampled from the frontal pole, the motor cortex was sampled along the medial longitudinal fissure caudal to the superior frontal gyrus, and the somatosensory cortex was collected caudal to the motor cortex at the ectomarginal gyrus, with the visual cortex collected at the occipital pole. The amygdala was identified in the temporal lobe, lateral of the optic chiasm, with the hippocampus caudal to the amygdala. The thalamus was collected from a mid-sagittal dissection superior of the hypothalamus. The hypothalamus was collected from the ventral surface caudal to the optic chiasm and rostral to the mammillary bodies. The head of the caudate was collected by removing the overlaying cortex. Ventral and lateral to the caudate lies the putamen, which was identified by a horizontal cut with the internal capsule separating the caudate and putamen. The medulla was collected at the caudal end. The midbrain tectum and tegmentum were collected at the level of the cerebral aqueduct. The cerebellum was collected lateral to the vermis on the superior surface. The pons was collected lateral to midline from the ventral surface inferior to the cerebellum and caudal to the midbrain.

### 2.3. Fatty Acid Analysis

Samples of each brain region were stored in sealed vials with air spaces cleared with a stream of nitrogen gas and stored at −80 °C until fatty acid analysis within 60 days. Forage and lick tub samples were obtained monthly during the 220 d of feeding and then composited. The forage samples were lyophilized, ground, and stored at −20 °C in sealed plastic bags. The lick tubs were sampled by drilling the cores to obtain a cross section, with the samples stored in sealed vials at −80 °C. For the brain fatty acid analysis, approximately 0.5 g of tissue was sub-sampled and weighed into 16 × 125 screw cap tubes and lyophilized for 5 d at −1 °C. The fatty acid methyl esters were prepared from the total lipids of each brain sample, in duplicate, by direct transesterification using 0.2 *M* KOH in methanol [28]. The fatty acid methyl esters of the forage total fatty acids were prepared by subjecting 0.5 g of dried forage to direct transesterification using 1.09 *N* HCL in methanol [29]. The total lipids of 0.5 g of the lick tub core samples were extracted in 7.6 mL of a mixture of chloroform, methanol, and water (1: 2: 0.8, vol/vol/vol) [30]. This extraction included a step in which the lipid-containing organic phase was separated from the aqueous phase by washing with 2.0 mL of a 1.0 *M* KCl plus 0.15 *N* HCl mixture. The decreased pH acidified the fatty acids of the calcium salts, ensuring their extraction into the organic phase. The organic chloroform phase was transferred to clean 16 × 125 tubes and dried under a stream of nitrogen gas, and the fatty acid residue was converted to fatty acid methyl esters by incubating in 2.0 mL of 1.09 *N* HCl for 60 min at 85 °C in a dry bath incubator.

All fatty acid methyl ester preparations were extracted in 2.0 mL of hexane and transferred into 2.0 mL gas-liquid chromatography autosampler vials and sealed. Each fatty acid methyl ester sample included 1.0 mg of methyl tridecanoate as an internal standard. The fatty acid composition was determined by using gas-liquid chromatography with an Agilent Technologies 6890 N GC equipped with an Agilent Technologies 7683 Automatic Sample Injector. The fatty acid methyl esters were separated using a 60 m × 0.25 mm (i.d.) DB-23 capillary column with a 0.25 µm film thickness (Agilent J&W Columns), with the injector and detector temperatures at 250 °C. The column oven temperature was maintained at 75 °C for 1 min and then increased to 170 °C at 6.5 °C/min and maintained for 27 min, followed by an increase at 10 °C/min to 215 °C. The fatty acids were identified by co-elution with purified standards (Nu-Chek Prep, Inc., Elysian, MN, USA). The fatty acid concentrations, mg fatty acid/100 mg of total fatty acids, were calculated from the peak areas integrated using ChemStation software (Agilent Technologies, Santa Clara, CA, USA).

The data were analyzed as a 2 × 2 factorial-designed experiment using the GLM procedure of SAS (SAS Institute Inc., ver. 9.3, Cary, NC, USA) to determine the effects of the supplement and sex on the fatty acid concentrations within and across the brain regions. The results are least squared means determined by the GLM procedure. The tissue concentrations of DHA were correlated to C18:1 n9, C16:0 + C18:0, and AA (C20:4 n6) using correlation procedures (PROC CORR) of SAS.

## 3. Results

The supplement disappearance equated to an average 0.02 kg/d/animal intake for both the fish and palm oil supplements. The serum and liver concentrations of the fatty acids did not differ by sex (*p* > 0.05) or sex by supplement interaction (*p* > 0.05). Largely reflecting the fatty acid profiles of the provided supplements (Table 1), serum concentrations of C18:0 (Stearic), C18:1 n-9 (Oleic), and C20:4 n-6 (AA) were greater (*p* < 0.01) in cattle provided with supplemental palm oil (Table 2). The liver fatty acid profiles were similar with greater (*p* < 0.01) concentrations of C18:1 n-9, C18:2 n-6 (Linoleic), and AA in the palm-oil-supplemented cattle (Table 2). Cattle provided with diets supplemented with the calcium salts of fish oil had greater (*p* < 0.01) serum and liver EPA and DHA contents, with increased (*p* < 0.01) liver contents of C18:3 n-3 (α Linolenic acid, ALA; Table 2).

The fatty acid contents of all the brain regions by supplementation can be accessed in Appendix A. Across all the brain regions measured, C16:0 (17.7% ± 0.4), C18:0 (20.4% ± 0.3), C18:1 n-9 (24.3% ± 0.8), C20:4 n-6 (5.7% ± 0.3), and C22:6 n3 (9.3% ± 0.5) were the predominant fatty acids, accounting for more than 75% of the fatty acid contents of the brain samples. The palmitic (C16:0), stearic (C18:0), and oleic (C18:1 n-9) fatty acids did not differ across the brain regions, other than in the midbrain tegmentum, where the fish-oil-supplemented cattle had greater (*p* < 0.05) contents of oleic acid than the palm-oil-supplemented cattle (31.81 vs. 30.63 ± 0.31 mg/100 mg, respectively). Across most areas of the brain, greater contents (*p* ≤ 0.05) of docosahexaenoic acid (DHA, C22:6 n3) were noted in the cattle supplemented with fish oil, while the cattle provided with supplements of palm oil had greater (*p* ≤ 0.05) concentrations of AA (C20:4 n-6; Figure 1). Differences among the supplemented cattle in the AA or DHA contents of the thalamus or hypothalamus were not detected (*p* > 0.05).

The fatty acids C18:1 n-7 (vaccenic), C20:1 n-9 (eicosenoic), C22:4 n-6 (adrenic), and C22:5 n-3 (docosapentaenoate) all occurred in the brains of the cattle at concentrations of greater than 1% but less than 5% of all the fatty acids. The remaining identified fatty acids (C14:0, C16:1 n-7, C17:0, C18:2 n-6, C18:3 n3, C20:0, C20:2 n-6, C20:3 n-6, C20:5 n-3 (eicosapentaenoic acid, EPA), C22:0, and C22:1 n-9) occurred at less than 1% of the total brain fatty acid contents. Across most brain regions, C22:4 n-6 concentrations were lower (*p* < 0.05) and EPA and C22:5 n-3 concentrations were greater (*p* < 0.05) in the cattle fed fish oil calcium salt supplementation compared to palm oil supplementation. The concentrations of vaccenic and eicosenoic were similar (*p* > 0.05) across most brain regions, with the palm-oil-supplemented cattle having greater (*p* ≤ 0.05) concentrations of vaccenic and eicosenoic acid in the parietal lobe and sensory cortex, respectively (Ca18:1 n-7, 4.58 vs. 4.37 ± 0.06; C20:1 n-9, 2.16 vs. 1.68 ± 0.12, respectively).

In the amygdala, significant interactions between sex and diet (*p* < 0.05) were observed for C20:2 n-6 and C22:4 n-6. The palm-oil-supplemented cattle had greater (*p* = 0.003) concentrations of C20:2 n-6 in steers than in heifers (1.15 vs. 0.68 ± 0.09, respectively), whereas C22:4 n-6 was greater (*p* = 0.003) in heifers than steers (6.32 vs. 5.49 ± 0.20, respectively). Sex by treatment interactions (*p* ≤ 0.05) were observed in the hippocampus for C18:0 and DHA. Heifers provided with fish oil supplements had greater (*p* = 0.02) concentrations of C18:0 (stearic acid) than similarly supplemented steers (21.05 vs. 20.34 ± 0.24, respectively). However, in the palm-oil-fed cattle, the steers had greater (*p* = 0.03) concentrations of DHA than the heifers (9.98 vs. 8.54 ± 0.03, respectively). In the midbrain tectum, a significant interaction (*p* = 0.04) was observed for C22:1 n-9 in the fish-oil-fed cattle, where the steers had greater concentrations than the heifers (0.72 vs. 0.50 ± 0.05, respectively).

In the midbrain tegmentum, several interactions between diet and sex were observed. For C16:0, a weak interaction (*p* = 0.09) occurred in which the palm-oil-supplemented heifers had greater (*p* = 0.03) concentrations than similarly supplemented steers (15.08 vs. 14.10 ± 0.30, respectively). For C17:0, the interaction (*p* = 0.05) revealed heifers with greater concentrations (*p* = 0.03) than steers, but only in those fed the fish oil supplement (0.61 vs. 0.56 ± 0.02, respectively). Significant interactions occurred for C20:2 n-6 (*p* = 0.02) and C22:4 n-6 (*p* = 0.01), while a weak interaction was observed for AA (*p* = 0.06). For the palm-oil-fed cattle, C20:2 n-6 concentrations were greater (*p* = 0.04) in steers than in heifers (0.92 vs. 0.70 ± 0.06, respectively). The palm-oil-fed heifers had greater concentrations (*p* = 0.003) of C22:4 n-6 than the steers fed palm oil (5.44 vs. 4.60 ± 0.17, respectively). The palm-oil-fed heifers also had greater (*p* = 0.03) AA concentrations than the palm-oil-fed steers (4.94 vs. 4.14 ± 0.25, respectively). In the medulla, the only significant interaction was observed for that for ALA (*p* = 0.05), in which the steers tended to have greater (*p* = 0.08) concentrations, but only for the palm-oil-supplemented cattle (0.53 vs. 0.41 ± 0.04). In the pons, an interaction was observed for C18:1 *cis*-11 in which the fish-oil-fed heifers had greater (*p* = 0.01) concentrations than the fish-oil-fed steers (4.67 vs. 4.21 ± 0.12, respectively).

For the internal capsules, significant interactions (*p* ≤ 0.06) were observed for C20:2 n-6 and 22:1 n-9, as was a weak interaction for C22:5 n-3. In the fish-oil-fed cattle, the heifers had greater (*p* ≤ 0.02) concentrations of C20:2 n-6 (1.04 vs. 0.69 ± 0.06, respectively) and C22:1 n-9 (1.90 vs. 1.34 ± 0.16, respectively) than the steers. For C22:5 n-3. the fish-oil-fed steers had greater concentrations than the fish-oil-fed heifers (1.44 vs. 1.11 ± 0.07, respectively).

The concentrations of DHA by brain region and tract in ascending concentrations (Figure 2A) ranged from approximately 3 mg of DHA per 100 mg of total fatty acids for the medulla to 13 mg for the visual cortex.

The change in concentration of C18:1 n-9 across the brain regions and tracts relative to DHA is shown in Figure 2B. With increasing concentrations of DHA, C18:1 n-9 concentrations decreased linearly across the brain regions and tracts, with an R^2^ value of 0.89, indicating a strong relationship between the changes in the proportions of these fatty acids in the bovine brain. When the concentrations of C16:0 and C18:0 were combined and plotted against DHA concentrations, a positive correlation was observed, with R^2^ = 0.85 (Figure 2C). The decrease in the concentrations of these two fatty acids combined (approximately 14 mg/100 mg) was similar to that for DHA (approximately 10 mg/100 mg).

The concentrations of AA did not change as consistently with the concentrations of DHA, but when plotted against the concentrations of DHA, an R^2^ value of 0.56 was observed (Figure 2D).

## 4. Discussion

The fatty acid compositions of wild ungulate (elk, deer, and pronghorn) homogenized brains [22] are consistent with the fatty acid compositions of bovine brains in the current study. The magnitude of the major saturated fatty acids (C16:0 and C18:0) and unsaturated fatty acids (C18:1 n-9, ALA, AA, EPA, and DHA) are consistent across the species. Prior to 1981, bovine brain lipid research was limited to evaluating lipid classes and fatty acids within lipid classes of whole brain homogenates [26]. The present study is novel in that it presents a comprehensive fatty acid profile for bovine brain.

Neural development requires DHA, which is stored in the neutral lipids of neuronal nerve growth cones for recruitment during neurite development [31]. Neurogenesis, neuroblast migration and differentiation, synaptic genesis, and axon myelination correlate with DHA in the brain [5]. In the mature brain, DHA is associated with reduced aggressive behavior [32]. Although behavior was not tracked in the current study, DHA comprises approximately 10% of the brain fatty acids in the bovine brain, with greater contents in the cortex than in the brainstem, making an influence on behavior seem possible. Small changes in dietary fatty acids influenced the fatty acid contents of neural and glial cell membranes from the frontal region, cerebellum, and hippocampus of rats [33]. Mature animals would be more resistant to changes in DHA supply to the brain, as evidenced by the greater half-life of brain DHA during ALA-reduced intake in rodents [5]. Dietary DHA was associated with greater regional brain volumes to include the hippocampus and amygdala, as well as improved cognitive measures [34]. How these effects extrapolate to behavior in cattle is unknown.

Rats fed soybean oil were better able to maintain body temperature, were more pain-tolerant, and had improved cognitive performances than lard-fed rats [35]. Mice supplemented with n-3 PUFA after a period of ALA-deficiency had increased n-3 PUFA in several brain regions, wherein their saturated fatty acid concentrations were not affected by diet [1]. Further, DHA supplementation in bottle-fed baboons did not result in consistent changes in specific brain regions, and AA supplementation did not influence brain AA to any extent [20]. In the present study, the levels of C22:4 n-6 and AA were consistently greater across the brain regions in the palm- vs. fish-oil-supplemented cattle. The fish-oil-supplemented cattle had greater DHA contents across the regions. Brain DHA concentrations in the present study were consistent with those reported by Diau et al. (2005), at approximately 10% of all the fatty acids across most regions. Arachidonic acid in the baboon brain also varied with region and tract such that the highest concentration was observed in the amygdala, followed by (in descending order of concentration): frontal lobe > putamen > caudate > hippocampus > hypothalamus > cerebellum > thalamus [20]. In the current study, bovine brain AA concentrations were similar to those reported for baboons, except for that of the frontal lobe, which was relatively low compared to those reported by Diau et al. (2005).

The bovine digestive tract limits the amount of PUFA obtained from dietary sources through the hydrogenation of carbon-carbon double bonds of these fatty acids in the rumen [24]. Moreover, the ruminant animal does not synthesize the n-6 or n-3 fatty acids de novo to serve as precursors for PUFAs, such as DHA or AA. Nevertheless, the accumulation of significant fatty acids in the brain requires dietary, as well as endogenous, supply. The two predominant brain PUFAs, DHA and AA, rely mainly on provision from the liver or dietary supply and not on synthesis by elongation and desaturation [8,36,37]. These authors also reported higher rates of the synthesis of DHA from ALA in the livers than in the brains of rats. Livers from the cattle in the present study contained both ALA and DHA, indicating that ALA from the forage diet and supplement is not completely reduced in the rumen. Since DHA can be synthesized from ALA, the hepatic supply of PUFAs for brain uptake is sustained.

While non-esterified fatty acids (NEFAs) diffuse through the blood–brain barrier [13,14,15], this source of DHA and AA uptake by the brain represents a small fraction of the brain’s fatty acid content [8]. Transport proteins also contribute to fatty acid uptake, especially during development, and then later, diffusion becomes more prevalent [2,15]. The long half-lives of DHA, AA, and C16:0 in brain tissue explains the slow turnover of these fatty acids and how plasma fatty acids contribute only a small fraction of the fatty acids observed in brain lipids [38]. The small diet effect on the bovine brain levels of these fatty acids in the current study is consistent with this model. The involvement of DHA and AA in signal transduction requires the hydrolysis of the phospholipid. Re-esterification to phospholipids results in their conservation and long half-lives [38]. In the present study, n-3 PUFAs were present in the sera and livers of the cattle fed either fish or palm oil calcium salts, with more DHA and EPA noted in the fish-oil-supplemented cattle.

Sex-specific differences in bovine brain fatty acids were noted in the low-abundance fatty acids. In the brain homogenates of female rats fed an ALA-deficient diet, brain DHA concentrations decreased after one reproductive cycle, with a commensurate increase in C22:5 n-6 that persisted through two further reproductive cycles [16]. Both male and ovariectomized female rats had less erythrocyte DHA than females supplemented with ALA; however, females had lower DHA in the prefrontal cortex than males when ALA supplementation was provided [19]. Lower total brain phospholipid-DHA was reported in male rates compared to female rats, which was associated with the decreased locomotor activity of the males [17]. This suggests that differences between male and female brain fatty acid composition could promote differences in behavior. In the present study, greater DHA in the hippocampus of steers than that in heifers was observed, but only for the cattle fed the palm oil supplement. However, the heifers had greater DHA concentrations in the midbrain tegmentum than the steers, regardless of the dietary supplement. Since the males in the present study were gonadectomized, the influence of testosterone on brain DHA levels in cattle remains to be determined.

Arachidonic acid is present at a substantial level in bovine and rat brains. Linoleic acid is the major precursor of AA, yet most of the C18:2 n-6 that crosses the blood–brain barrier is metabolized by beta-oxidation, after which C16:0 and C18:0 fatty acid synthesis occurs [9]. These authors reported lipases with greater preferences for C18:2 n-6 than AA in brain esterified lipids, and they conclude the rate of AA uptake was equal to the rate of loss. The rapid loss of C18:2 n-6 and ALA by brain tissue is likely a consequence of the rich mitochondrial concentration at the blood–brain barrier [10].

The level of brain EPA observed in the present study (<1%) and that reported for other species are quite low considering that the serum concentrations of EPA in the cattle fed the fish oil calcium salts were approximately 9%. In the mouse brain, a greater retention of DHA than EPA was observed due to the rapid beta-oxidation of EPA [11]. In addition to the rapid beta-oxidation, decreased incorporation, elongation to DPA, and lower phospholipid recycling contributed to low brain EPA levels [39]. In the present study, fish oil supplementation increased brain EPA concentrations in cattle. Although the increases were significant, the levels remained low.

## 5. Conclusions

This study provided fatty acid compositions for various regions of the bovine brain, with major fatty acids at similar magnitudes as have been reported for other species. This indicates that the diversity of species for which brain fatty acids have been studied have similar, and therefore conserved, aspects of lipid metabolism, and it creates a fairly uniform brain fatty acid profile.

## Figures and Tables

**Figure 1 animals-12-02696-f001:**
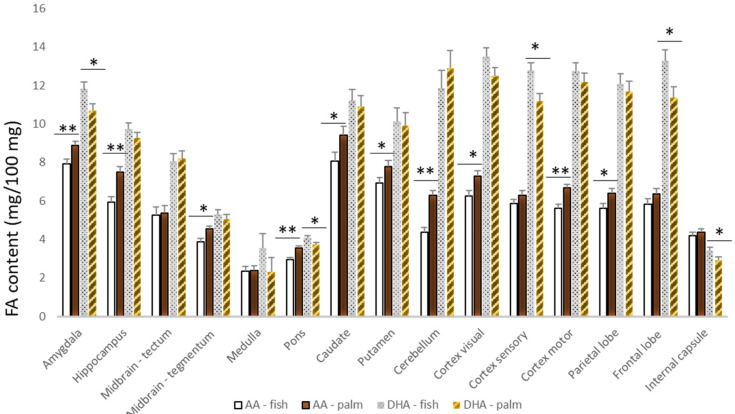
Content (mg/100 mg) of arachidonic acid (AA) and DHA across the brain regions of the cattle supplemented with the calcium salts of fish or palm oil (* *p* ≤ 0.05; ** *p* ≤ 0.01).

**Figure 2 animals-12-02696-f002:**
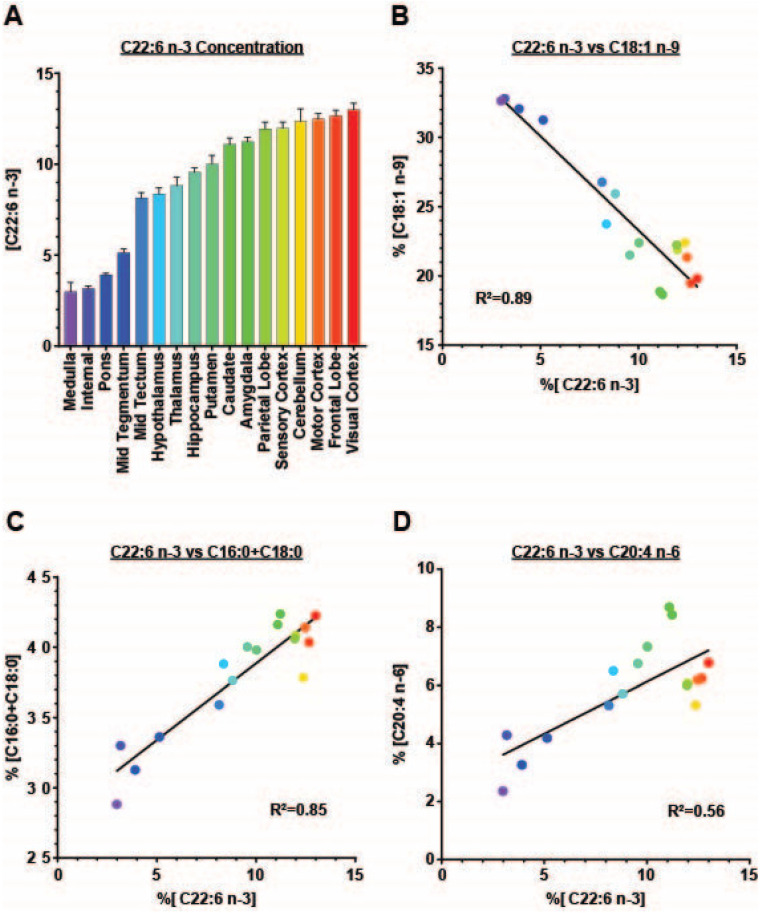
(**A**) Average concentrations of DHA (C22:6 n3) across the brain regions, and the correlation between DHA (C22:6 n3) and 18:1 n9 (**B**), C16:0 + C18:0 (**C**), and arachidonic acid (C20:4 n6) (**D**).

**Table 1 animals-12-02696-t001:** Fatty acid compositions (mg fatty acid/100 mg total fatty acids) of the forage diet and the fat supplements.

Fatty Acid	Forage	Fish Oil Calcium Salt	Palm Oil Calcium Salt
C14:0	0.90	6.08	0.81
C16:0	24.20	21.08	48.76
C16:1	2.13	6.39	1.92
C18:0	3.29	6.89	3.74
C18:1 n-9	2.22	14.99	34.11
C18:2 n-6	18.88	6.25	10.15
C18:3 n-3	48.38	1.06	0.51
C20:4 n-6	--	1.13	--
C20:5 n-3 (EPA)	--	11.23	--
C22:5 n-3	--	1.80	--
C22:6 n-3 (DHA)	--	7.73	--

**Table 2 animals-12-02696-t002:** Concentrations (mg fatty acid/100 mg of total fatty acids) of the total fatty acids of the sera and livers of the cattle that were group-fed forage with supplemental dried molasses lick tubs that contained 30% (by weight) calcium salts of either fish oil or palm oil fatty acids.

	Serum Fatty Acids			Liver Fatty Acids		
Fatty Acid ^a^	Fish Oil	Palm Oil	SEM ^b^	Fish Oil	Palm Oil	SEM ^b^
C16:0	13.39	14.01	0.31	15.00	16.00	0.59
C18:0	15.93	17.70	0.32 **	28.00	27.40	0.50
C18:1 n-9	6.24	9.18	0.20 **	7.94	10.87	0.45 **
C18:2 n-6	24.39	27.87	1.30	5.69	6.61	0.19 **
C18:3 n-3	10.96	9.69	0.47	1.98	1.66	0.07 **
C20:4 n-6 (AA)	3.01	3.92	0.20 **	4.98	7.80	0.13 **
C20:5 n-3	8.86	2.06	0.46 **	6.30	2.52	0.25 **
C22:6 n-3 (DHA)	2.27	1.08	0.20 **	6.55	2.72	0.22 **

^a^ refers to the number of carbon atoms: number of carbon-carbon double bonds. The “n” indicates the number of carbon atoms removed from the methyl end to the first carbon-carbon double bond. ^b^ refers to the standard error of the mean for 10 cattle (five heifers and five steers) per treatment. ** *p* < 0.01.

## Data Availability

The data available on request.

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
