# Peer review of "Dietary Fatty Acid Composition Impacts the Fatty Acid Profiles of Different Regions of the Bovine Brain"

_animals, 2022, doi:10.3390/ani12192696_

Round 1

Reviewer 1 Report

Dear authors,

The manuscript titled 'Dietary fatty acid composition IMPACTS fatty acid profiles of different regions of the bovine brain' is addressing an interesting topic of research for publication at Animals MDPI. See below a list of  comments to be reviewed before acceptance of the article for publication.

L1 Replace 'IMPACTS' by 'impacts' without using capital letters.

L122 Replace '-80 Cº' by '-80ºC'.

L124 Replace '-20 Cº' by '-20ºC'.

L126 Replace '-80 Cº' by '-80ºC' and delete an space before 'For brain ...'.

L128 Replace '-1.0 Cº' by '-1.0ºC'.

L139 Replace '-85 Cº' by '-85ºC'.

L147 Replace '250 º C' by '250ºC' and '75 º C' by '75ºC'.  

L148 Replace '170 º C' by '170ºC' and '6.5 º C' by '6.5ºC'.  

L149 Replace '10 º C' by '10ºC' and '215 º C' by '215ºC'.

L176 Replace 'vs' by 'vs.' in italics and delete '[ ]'.

L198 Replace 'Ca8:1 n-7' by 'C18:1 n-7' and 'vs' by 'vs.' in italics. Delete '[ ]'.

L204  Replace 'vs' by 'vs.' in italics and delete '[ ]'.

L205  Replace 'vs' by 'vs.' in italics and delete '[ ]'.

L208  Replace 'vs' by 'vs.' in italics and delete '[ ]'. Delete also an space before 'However, ...'.

L209  Replace 'vs' by 'vs.' in italics and delete '[ ]'.

L211  Replace 'vs' by 'vs.' in italics and delete '[ ]'.

L214  Replace 'vs' by 'vs.' in italics and delete '[ ]'.

L216  Replace 'vs' by 'vs.' in italics and delete '[ ]'.

L219  Replace 'vs' by 'vs.' in italics and delete '[ ]'.

L220 Replace 'vs' by 'vs.' in italics and delete '[ ]'.

L221 Replace 'vs' by 'vs.' in italics and delete '[ ]'.

L223 Replace 'vs' by 'vs.' in italics and delete '[ ]'.

L225 Replace 'vs' by 'vs.' in italics and delete '[ ]'.

L228 Replace 'vs' by 'vs.' in italics and delete '[ ]'.

L229 Replace 'vs' by 'vs.' in italics and delete '[ ]'.

L230 Replace 'vs' by 'vs.' in italics and delete '[ ]'.

L245 Replace 'Correlation of ...' by 'Correlation between DHA (C22:6 n-3) and C18:1 n-9 (B), C16:0-C18:0 (C) and arachidonic acid (C20:4 n-6) (D).

L262-263 Replace 'cer-ebellum' by 'ce-rebellum'.

L275 Replace 'vs' by 'vs.' in italics.

L280 Delete an space before '> caudate'.

L340-343 Delete this sentence and reformulate conclusions. Avoid to make any conjecture without any scientific support. Conclusions need to be supported by data and statements be justified by research conducted.

Best regards,

Reviewer.

Author Response

L1 Replace 'IMPACTS' by 'impacts' without using capital letters.

            Changed as suggested

L122 Replace '-80 Cº' by '-80ºC'.

            Changed as suggested

L124 Replace '-20 Cº' by '-20ºC'.      

            Changed as suggested

L126 Replace '-80 Cº' by '-80ºC' and delete an space before 'For brain ...'.

            Changed as suggested

L128 Replace '-1.0 Cº' by '-1.0ºC'.

            Changed as suggested

L139 Replace '-85 Cº' by '-85ºC'.

            Changed as suggested

L147 Replace '250 º C' by '250ºC' and '75 º C' by '75ºC'.  

            Changed as suggested

L148 Replace '170 º C' by '170ºC' and '6.5 º C' by '6.5ºC'.  

            Changed as suggested

L149 Replace '10 º C' by '10ºC' and '215 º C' by '215ºC'.

            Changed as suggested

L176 Replace 'vs' by 'vs.' in italics and delete '[ ]'.

            Changed as suggested

L198 Replace 'Ca8:1 n-7' by 'C18:1 n-7' and 'vs' by 'vs.' in italics. Delete '[ ]'.

            Changed as suggested

L204  Replace 'vs' by 'vs.' in italics and delete '[ ]'.

            Changed as suggested

L205  Replace 'vs' by 'vs.' in italics and delete '[ ]'.

            Changed as suggested

L208  Replace 'vs' by 'vs.' in italics and delete '[ ]'. Delete also an space before 'However, ...'.

            Changed as suggested

L209  Replace 'vs' by 'vs.' in italics and delete '[ ]'.

            Changed as suggested

L211  Replace 'vs' by 'vs.' in italics and delete '[ ]'.

            Changed as suggested

L214  Replace 'vs' by 'vs.' in italics and delete '[ ]'.

            Changed as suggested

L216  Replace 'vs' by 'vs.' in italics and delete '[ ]'.

            Changed as suggested

L219  Replace 'vs' by 'vs.' in italics and delete '[ ]'.

            Changed as suggested

L220 Replace 'vs' by 'vs.' in italics and delete '[ ]'.

            Changed as suggested

L221 Replace 'vs' by 'vs.' in italics and delete '[ ]'.

            Changed as suggested

L223 Replace 'vs' by 'vs.' in italics and delete '[ ]'.

            Changed as suggested

L225 Replace 'vs' by 'vs.' in italics and delete '[ ]'.

            Changed as suggested

L228 Replace 'vs' by 'vs.' in italics and delete '[ ]'.

            Changed as suggested

L229 Replace 'vs' by 'vs.' in italics and delete '[ ]'.

            Changed as suggested

L230 Replace 'vs' by 'vs.' in italics and delete '[ ]'.

            Changed as suggested

L245 Replace 'Correlation of ...' by 'Correlation between DHA (C22:6 n-3) and C18:1 n-9 (B), C16:0-C18:0 (C) and arachidonic acid (C20:4 n-6) (D).

            Changed as suggested

L262-263 Replace 'cer-ebellum' by 'ce-rebellum'.

            The computer is hyphenating the word. It is different now due to the changes.

L275 Replace 'vs' by 'vs.' in italics.

            Changed as suggested

L280 Delete an space before '> caudate'.

            Changed as suggested

L340-343 Delete this sentence and reformulate conclusions. Avoid to make any conjecture without any scientific support. Conclusions need to be supported by data and statements be justified by research conducted.

            Agreed, and changed as suggested.

Reviewer 2 Report

Authors performed a novel FA concentrations in bovine brain. It has been properly conducted and described. I have some minor comments.

Some minor comments:

- ln 2. why cappital letters?

- ln 48. AA?? the first time you mention an acronym, please explain

- ln 85. Include section number

- ln 87. Number or reference of the ethical committee aproval

- ln 89. provide more information about the supplements

- ln 96. include age range

- ln 101. avoid "killed", use slaughtered or something similar

- ln 152. delete ","

- table 1. I would include this in M&M when providing the supplements information, because this is not a result of the study...

- table 2. avoid that wide black line

- ln 188-192. These data could be shown in a suppl table, because it would be helpful for future studies in this area

- figure 1. Improve this figure in order to be more understandable: mainly increase size

- same comment for figure 2 than previously did for 1

- missing more correlation results (maybe include a table)

Author Response

- ln 2. why cappital letters?

            This was an error. It has been made lower case.

- ln 48. AA?? the first time you mention an acronym, please explain

            AA is defined in its first use in the body of the manuscript on line 47.

- ln 85. Include section number

            Section numbers have been added throughout the methods section.

- ln 87. Number or reference of the ethical committee approval

            IACUC approval number is now included in the body of the manuscript

- ln 89. provide more information about the supplements

Additional information is now included in the manuscript including the manufacturer of the fatty acid calcium salts, the formulator of the supplement, and intake.

- ln 96. include age range

            Age of cattle at the start and at collection has now been included.

- ln 101. avoid "killed", use slaughtered or something similar

There is just not a better word “slaughtered” is a different process. Killed is the correct term.

From Merriam-Webster dictionary: Killed (verb): cause the death of (a person, animal, or other living thing)

- ln 152. delete ","

            Changed as suggested

- table 1. I would include this in M&M when providing the supplements information, because this is not a result of the study...

            Moved to the Methods section as suggested

- table 2. avoid that wide black line

            Changed as suggested

- ln 188-192. These data could be shown in a suppl table, because it would be helpful for future studies in this area

            I think this is a great idea. Complete data is now included in supplemental tables 1 – 6.

- figure 1. Improve this figure in order to be more understandable: mainly increase size

            Increased size as suggested.

- same comment for figure 2 than previously did for 1

            Increased size as suggested

- missing more correlation results (maybe include a table)

Not all fatty acids were correlated. Only targeted and meaningful correlations. Correlations listed in the methods section are in the results.